# Provision of essential evidence-based interventions during facility-based childbirth: cross-sectional observations of births in northeast Nigeria

Josephine Exley [1], Claudia Hanson,[1,2] Nasir Umar,[1] Barbara Willey,[3] Abdulrahman Shuaibu,[4] Tanya Marchant [1]

¹Department of Disease Control, London School of Hygiene and Tropical Medicine, London, UK
²Public Health Sciences, Karolinska Institute, Stockholm, Sweden
³Department of Infectious Disease Epidemiology, London School of Hygiene and Tropical Medicine, London, UK
⁴The Executive Secretary, Gombe State Primary Health Care Development Agency, Gombe, Nigeria

**Correspondence to**
Ms Josephine Exley;
Josephine.Exley@lshtm.ac.uk

## ABSTRACT

**Objectives** To measure the provision of evidence-based preventive and promotive interventions to women, and subsequently their newborns, during childbirth in a high-mortality setting.

**Design and participants** Cross-sectional observations of care provided to women, and their newborns during the intrapartum and immediate postpartum period using a standardised checklist capturing healthcare worker behaviours regarding lifesaving and respectful care.

**Setting** Ten primary healthcare facilities in Gombe state, northeast Nigeria. The northeast region of Nigeria has some of the highest maternal and newborn death rates globally.

**Main outcome measures** Data on 50 measures of internationally recommended evidence-based interventions and good practice.

**Results** 1875 women were admitted to a health facility during the observation period; of these, 1804 gave birth in the facility and did not experience an adverse event or death. Many clinical interventions around the time of birth were routinely implemented, including provision of uterotonic (96% (95% CI 93% to 98%)), whereas risk-assessment measures, such as history-taking or checking vital signs were rarely completed: just 2% (95% CI 2% to 7%) of women had their temperature taken and 12% (95% CI 9% to 16%) were asked about complications during the pregnancy.

**Conclusions** The majority of women did not receive the recommended routine processes of childbirth care they and their newborns needed to benefit from their choice to deliver in a health facility. In particular, few benefited from even basic risk assessments, leading to missed opportunities to identify risks. To continue with the recommendation of childbirth care in primary healthcare facilities in high mortality settings like Gombe, it is crucial that birth attendant capacity, capability and prioritisation processes are addressed.

## INTRODUCTION

Global efforts to reduce preventable maternal and newborn mortality have focused on skilled attendance at birth. This has resulted in marked increases in coverage of births with a skilled birth attendant, mainly operationalised through childbirth in facilities. However, these increases have not been accompanied by the anticipated improvements in maternal and newborn outcomes in many low-income and middle-income countries,[1–4] prompting a closer examination of the quality of care provided.[5–7] A growing body of evidence from low-income settings highlights low provider skills and limited facility capability to provide good-quality care at birth.[1 8–10] A recent review found large declines in the proportion of individuals estimated to have skilled birth attendance when some measure of quality was taken into account.[11]

Good quality of care includes the timely and appropriate use of evidence-based clinical and non-clinical interventions that are acceptable to women.[12–14] WHO has developed evidence-based guidelines around intrapartum care, which have the potential to support healthcare providers to identify gaps in the quality of care and improve the

provision and experience of care.[15] A continuing challenge, however, is that detailed evidence on the extent to which the recommended interventions are practised during routine childbirth in health facilities in low-income countries is scarce.[1][16] Available studies have predominantly focused on readiness for quality such as availability of drugs, supplies and other inputs, while neglecting processes of care.[16] In part, this is because these data are made available through surveys similar to the Service Provision Assessments,[17] or through routine health information systems—information sources that include very little process quality data. Household surveys, including the Demographic and Health Survey and the Multiple Indicator Cluster Survey, do not collect extensive data on the content of care during childbirth, in part because evidence suggests women's self-reports have low validity.[18][19] This lack of data on coverage of evidence-based care during childbirth impedes decision making around possible solutions. Although not a large-scale measurement solution, observations provide useful insight in to the behaviours of healthcare workers and could support strategies to improve care.

In this study of birth observations, we aimed to examine the maternal and newborn outcomes experienced by all women admitted for childbirth and postpartum haemorrhage in a sample of primary healthcare facilities in Gombe state, Nigeria. For women who had an uncomplicated labour, we evaluated the provision of evidence-based care provided to women, and subsequently their newborns, from initial assessment up to 1 hour post partum.

## METHODS

We conducted direct observations of childbirth care in 10 primary healthcare facilities, in Gombe state, Nigeria, approximately every 6 months over a 2-year period between June 2016 and August 2018.

### Study setting

Gombe state is one of six states in northeast Nigeria, it has an area of 20 265 km$^2$ and a population of 2 857 042.[20] Over 80% of the population live in rural areas and are reliant on subsistence farming as their primary source of income.[21] The northeast region of Nigeria has some of the highest maternal and newborn death rates globally, estimated at 1549 per 100 000 live births in 2015 and 33 per 1000 live births in 2017, respectively.[22][23]

Access to maternal healthcare services is relatively low in Gombe state. In 2018, 46% of women in the state reported at least one antenatal care visit from a doctor, nurse, midwife or nurse/midwife and 28% delivered in a health facility.[24] Over 70% of facility deliveries, in 2018, took place in a rural primary healthcare facility.[25] Recent work in Gombe on the drivers of attending a facility for childbirth found that health system conditions including availability of staff, drugs and supply, and a clean environment had the biggest influence on respondents' decision around where to give birth.[26]

Healthcare is predominantly delivered via a network of rural primary healthcare clinics run by the Gombe State Primary Healthcare Development Agency (GSPHCDA). In 2017, 460 primary healthcare clinics and 26 referral facilities provided childbirth services.[27] In primary healthcare facilities care is typically delivered by lower cadres of healthcare workers, for example, community health extension workers (CHEWS), junior CHEWS and health officers.[28][29] In response to the shortage and uneven distribution of healthcare workers, under its task-shifting and task-sharing policy for essential healthcare services, Nigeria classifies CHEWs as skilled birth attendants.[30]

Primary healthcare facilities in Gombe are poorly resourced, often lacking essential supplies and commodities to provide basic maternal and newborn healthcare.[31–33] Led by the GSPHCDA, since 2016 intense non-governmental organization activity has been ongoing in 57 primary healthcare facilities across Gombe state, aimed at increasing the quality of care.[34][35] Interventions include training of CHEWs in all aspects of skilled birth attendance and basic emergency obstetric care, and improving the supply of essential maternal and newborn health commodities.[36] These facilities provide basic emergency obstetric and newborn care. Emergency care and complicated cases from these health facilities are referred to referral facilities. None of the 57 primary healthcare facilities have a medical doctor, 4% have at least one nurse and 19% have at least one midwife.[37]

Sampling methods have been described in detail elsewhere.[19][32] Briefly, in November 2015, 10 primary healthcare facilities were selected from the 57 facilities for an in-depth assessment of quality of care. To achieve a sufficiently large number of observations and minimise the duration of data collection, the 10 primary healthcare facilities with the highest number of births in the preceding 6 months, as recorded in the maternity register, were purposively selected. The mean number of births per month in the 10 primary healthcare facilities was 15.7 (SD 12.0), compared with 4.3 (SD 6.3) births per facility per month across Gombe state as a whole.[19]

### Data collection

Five rounds of data collection took place over the 2-year study period. Each round lasted 3 weeks, during which observers aimed to collect data from a total of around 350 women. Two trained female observers (local midwives, not employed by the facility) and one clinical supervisor were assigned to each facility. Observers worked in 8 or 12 hours shifts to provide near continual data collection during the period. Depending on the observation team's work schedule, the first point of contact for any observation may have been during initial assessment of a newly admitted pregnant woman or at a later stage of labour. Observers aimed to observe all women who were admitted irrespective of the cadre of the attending healthcare worker, but they prioritised observing women during

the second and third stage of labour and immediately post partum rather than observing women earlier in the process. Observers stayed continuously with women from the first point of contact until the first hour after birth. The healthcare worker observed may have been different at different timepoints in the same facility. The clinical supervisor was always available onsite but not present in the delivery room.

A structured clinical observation checklist, administered on a Lenovo A3300 tablet using CSPro V.7.0 (US Census Bureau and ICF Macro, Suitland, Maryland, USA), was used to record the processes of care and birth attendant–client interactions and client characteristics. The content of the checklist was developed from the United States Agency for International Development (USAID)-funded Maternal and Child Health Integrated Programme's tool for observing vaginal births and the following complications: postpartum haemorrhage, pre-eclampsia/eclampsia and newborn asphyxia.[38] The checklist was piloted and modified to the Gombe context.

All women attending the facility in active labour or experiencing postpartum haemorrhage were invited to participate at the time of admission. All potential participants were provided with a study information sheet and a consent form in English and Hausa. Taking care to include any support persons accompanying potential participants, the observer read the information sheet, explained the purpose of the study, the risks and benefits of participating and answered questions before seeking written consent from the woman and verbal consent from the healthcare worker attending. Women who were not able to write their name were asked to provide a thumb print on the consent form. Participation was voluntary and participants were free to withdraw at any time.

Before each round of data collection, observers underwent 4 days of training on how to conduct unobtrusive observations, the safety and confidentiality protocols and how to ensure consistency of rating between observers. Throughout the observation period, clinical supervisors conducted spot checks of observers and data to provide ongoing quality assurance.

Observers were required to prioritise the safety of the mother and newborn; protocols were established on the actions to take during any life-threatening events. This included immediately stopping the observation activity and calling for the clinical supervisor who could advise the attending healthcare worker. A formal report detailing any actions and decisions made was made available to the Executive Secretary of the GSPHCDA. Where data collection was stopped, observations were excluded from the study.

### Defining provision of evidence-based care

For this analysis, the content of the clinical observation checklist was mapped against current recommendations for high quality mother and newborn care.[13 15 39–42] Fifty measures were identified (box 1), grouped into four organising categories based on the stage of childbirth:

---

**Box 1    Measures of evidence based childbirth care included in this analysis**

**History taking and initial assessment**
► Checks client card or asks client her age, length of pregnancy and parity.
► Asks whether woman has experienced any complications during current pregnancy.
► If woman has had any previous pregnancies, asks about complications during previous pregnancies.
► Checks client card or asks client her HIV status.
► Washes his/her hands with soap and water or uses disinfectant before initial any examination.
► Takes temperature.
► Takes blood pressure.
► Checks fetal heart rate with fetoscope/doppler/ultrasound.
► Performs vaginal examination.
► Encourages the women to have a support person present during labour and birth.
► Explains procedures to woman (support person) before proceeding.
► Asks woman (and support person) if she has any questions.

**First stage of labour**
► Partograph used to monitor labour.
► Washes his/her hands with soap and water or uses antiseptic prior to any examination of woman.
► Wears high-level disinfected or sterile surgical gloves.
► A support person (or companion) for mother is present at some point during labour.
► At least once, explains what will happen in labour to woman (support person).
► At least once, encourages woman to consume fluids/food during labour.
► Drapes woman (one drape under buttocks, one over abdomen).
► At least once, encourages/assists woman to ambulate and assume different positions during labour.
► Following items of equipment laid out in preparation for birth:
  – At least two cloths/blankets (one to dry; one to cover).
  – Sterile scissors or blade.
  – Disposable cord ties or clamps.
  – Suction bulb.
  – Newborn face mask size 0 or one and self-inflating ventilation bag (250 or 500 mL).

**Second and third stage of labour**
► Assisted by more than one healthcare worker at one point during labour.
► Woman gave birth in lithotomy position (on back).
► As baby's head is delivered, supports perineum.
► Administers uterotonic.
► Timing of uterotonic.
► If care provided by a skilled birth attendant, applies traction to the cord while applying suprapubic counter traction.
► Assesses completeness of the placenta and membranes.
► Assesses for perineal and vaginal lacerations.
► A support person (companion) for mother is present at birth.

**Immediate newborn and postpartum care**
► Drying baby immediately after birth with towel.
► Baby placed skin to skin with mother.
► Bathing delayed for at least 1 hour.

Continued

---

## Box 1    Continued

- ► Ties or clamps cord when pulsations stop, or by 2–3 min after birth (not immediately after birth).
- ► Cuts cord with clean blade or clean scissors.
- ► Breast feeding initiated within first hour.
- ► Takes mother's vital signs within the first 15 min after birth.
- ► Administers antibiotics to mother post partum.
- ► Checks baby's temperature within the first 15 min after birth.
- ► Baby weighed.
- ► Mother and newborn kept in same room after birth (rooming-in).
- ► Baby kept skin to skin for first hour.
- ► Administers vitamin K to newborn.
- ► Provides tetracycline eye ointment prophylaxis.
- ► Administers chlorhexidine to the newborn cord.
- ► If the mother is HIV positive, administers antiretroviral therapy to newborn.

(1) initial assessment; (2) first stage of labour; (3) second and third stage of labour and (4) immediate newborn and postpartum care.

### Inclusion criteria

Data from the five data collection periods were combined into a single dataset. Observations were excluded from the dataset if the woman's outcome was not recorded. For all women observed, we mapped the different pathways from admission to the facility (childbirth or postpartum haemorrhage event) to their outcome. For women who experienced an uncomplicated labour the outcome of their baby was also mapped. An uncomplicated labour was defined as a woman who was sent to the ward for recuperation or discharged home after birth and who did not experience an adverse event to her own health (referral, postpartum haemorrhage or pre-eclampsia/eclampsia) or death.

For the analysis of the provision of essential evidence based care, our population of interest was women with an uncomplicated labour and detailed information on their care and that of their newborn are included here. Women who were admitted but experienced an adverse event or death were excluded from the analysis because of their individual medical needs. For measures related to newborn care the analysis was further restricted to newborns recorded as being alive and who did not require resuscitation care or were not referred to another facility.

### Analysis

For each measure, per cent frequencies and 95% CIs were calculated, adjusted for clustering by primary healthcare facility and stratified by time point using the svyset and svy commands in STATA V.15.1 (StataCorp). Results are presented graphically by time point to highlight any variability and the average across all five time points is presented in the text.

### Patient and public involvement

Patients and the public were not involved in the design, conduct, reporting or dissemination plans of our research. Observations were recorded in English and pre-testing completed in health facilities by staff.

### RESULTS

In total 1875 women were admitted to a facility during the five observations periods. The median age of the women was 24 years (IQR 20–29) and for 19% it was their first birth: median parity 2 (IQR 1–5). The median gestational age of women on admission was 39 weeks (IQR 38–39); 6% of women had a gestational age of less than 37 weeks and for 21% of women gestational age was not recorded on their client card and/or they did not know. At the start of the observation period, 10% of women were attended by a skilled birth attendant (doctor, midwife, nurse), 15% by a CHEW, 25% by a junior CHEW and 50% by an 'other' birth attendant. 'Other' included environmental health officers/technicians/assistants (43%), health attendant/assistant (43%), traditional birth attendants (4%), community health officer (1%) and other (9%).

We first present the outcomes for all 1875 women who were admitted to the facility, followed by the coverage of the provision of evidence based care measures, outlined in box 1, for the 1804 women who had an uncomplicated labour and the 1635 babies born to these women who did not experience an adverse event or death. The full table of results disaggregated by time point is presented in online supplemental material.

### Outcomes

Figure 1 presents the outcomes for the 1875 women admitted to a facility during the five observation periods. Nine women were admitted with postpartum haemorrhage and 1866 were admitted for childbirth. Fourteen women admitted for childbirth were referred to another facility during labour. The main reason for referral was prolonged labour; three women were referred with severe pre-eclampsia and one had eclampsia. For the four women diagnosed with severe pre-eclampsia/eclampsia during labour, insufficient assessment and monitoring of the woman was noted by observers: 'there were delays in needed treatment. No observations undertaken during the first hour of admission to detect disease condition and take appropriate action', while another observed 'woman was admitted for normal labour but only vaginal examination was observed without physical nor vital signs or urine test'.

Of the 1852 women who gave birth in the facility, all delivered vaginally with one woman recorded as requiring an assisted delivery; caesarean sections are not available in primary healthcare facilities in this setting. Over half of births occurred on a weekday (56% (1031/1852]) between 7:00 and 19:00 hours (58% (1067/1852)). Two per cent (34/1852) of deliveries were multiple births.

Post partum, 45 women experienced an adverse event and three women died: the mortality rate was 1.6 per 1000

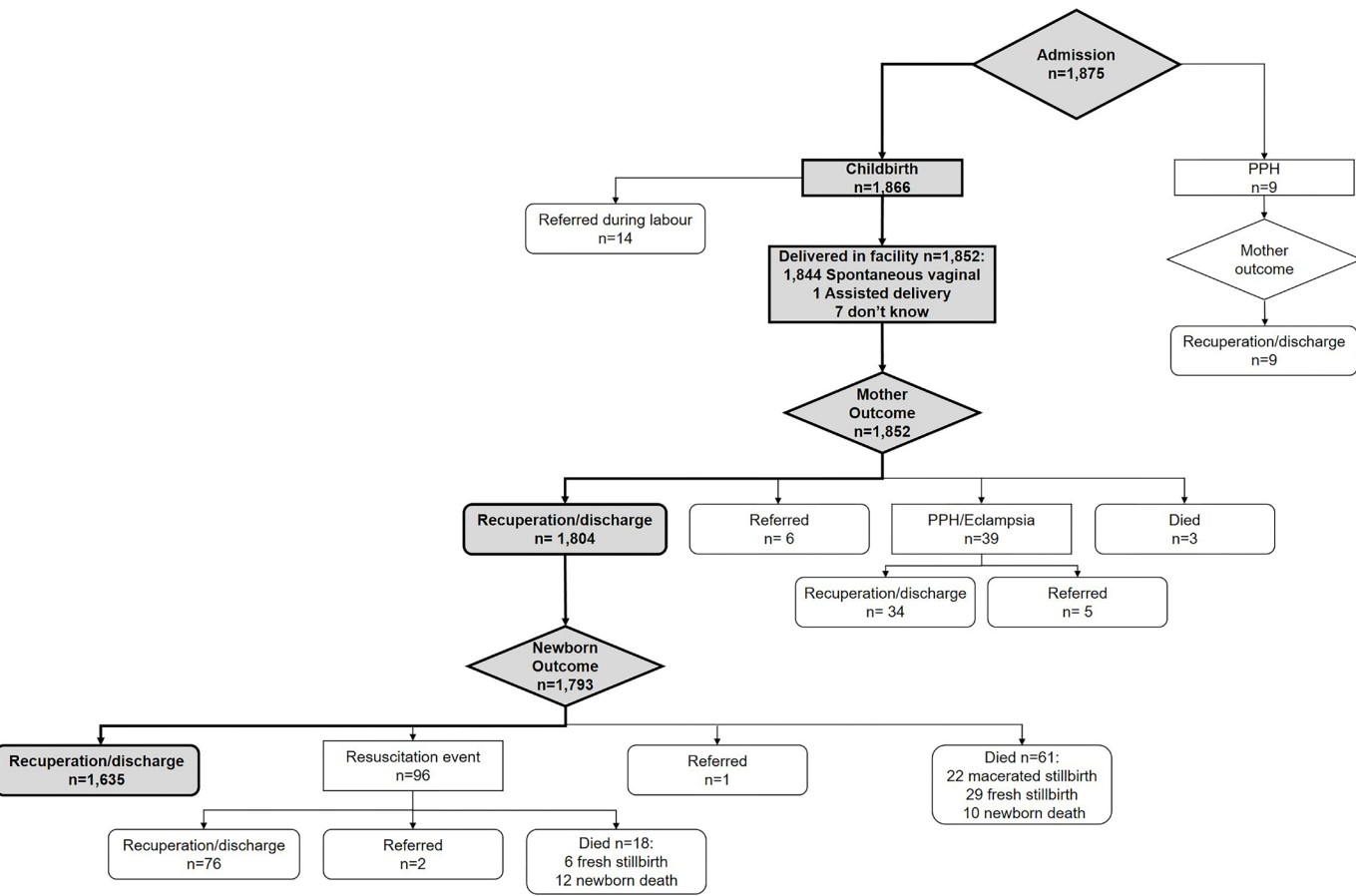

**Figure 1**  Women and newborn's pathway from admission. PPH, postpartum haemorrhage.

women. Five women were referred intrapartum due to complications associated with multiple births and six women were referred postpartum. Of the six women referred post partum, two women with eclampsia were referred because their condition could not be stabilised. Both women experienced convulsions and lost consciousness. They both received magnesium sulphate before referral: one woman's baby was born alive and referred with the mother, the other was a fresh stillbirth. Three women were referred following a postpartum haemorrhage event; all received oxytocin in the facility before referral and none had severe postpartum haemorrhage (abnormal bleeding of more than 1000 mL): one woman's condition was reported not to be stable and the baby required resuscitation before referral. The other two women were reported to be stable and their baby was referred with them. The remaining woman referred postpartum was referred because 'mother had a retain placenta, mentor and birth attendant tried everything but it failed'. The newborn was referred with their mother.

For the babies born to women who had an uncomplicated labour, 96 received resuscitation care of whom 19% subsequently died. Overall, 79 babies died: the perinatal mortality rate was 44.1 per 1000 newborns (figure 1). For babies born to women who experienced an adverse event or died, 19% (9/48) were referred and four died: the perinatal morality rate among this group was 83.3 per 1000 newborns.

## Provision of evidence-based care

All 1804 women who had an uncomplicated labour were observed during the second and third stage of labour. Observers were required to prioritise the second and third stage of labour so the number of women and newborns observed during other stages of childbirth varied (table 1).

### History taking and initial assessment

In total, 1801 women were observed during the initial assessment and history taking. Birth attendants were routinely observed to check the women's record for or,

**Table 1**  Proportion of all women admitted for childbirth who were observed, disaggregated by stage of care

|  | Percentage observed | |
| --- | --- | --- |
| Stage of care | Women | Newborns |
| Initial assessment and history taking | 99.8 (1801/1804) | n/a |
| First stage of labour | 59.3 (1069/1804) | |
| Second and third stage of labour | 100 (1804/1804) | |
| Immediate newborn and postpartum care | 94.1 (1697/1804) | 98.9 (1617/1635) |

n/a, not applicable.

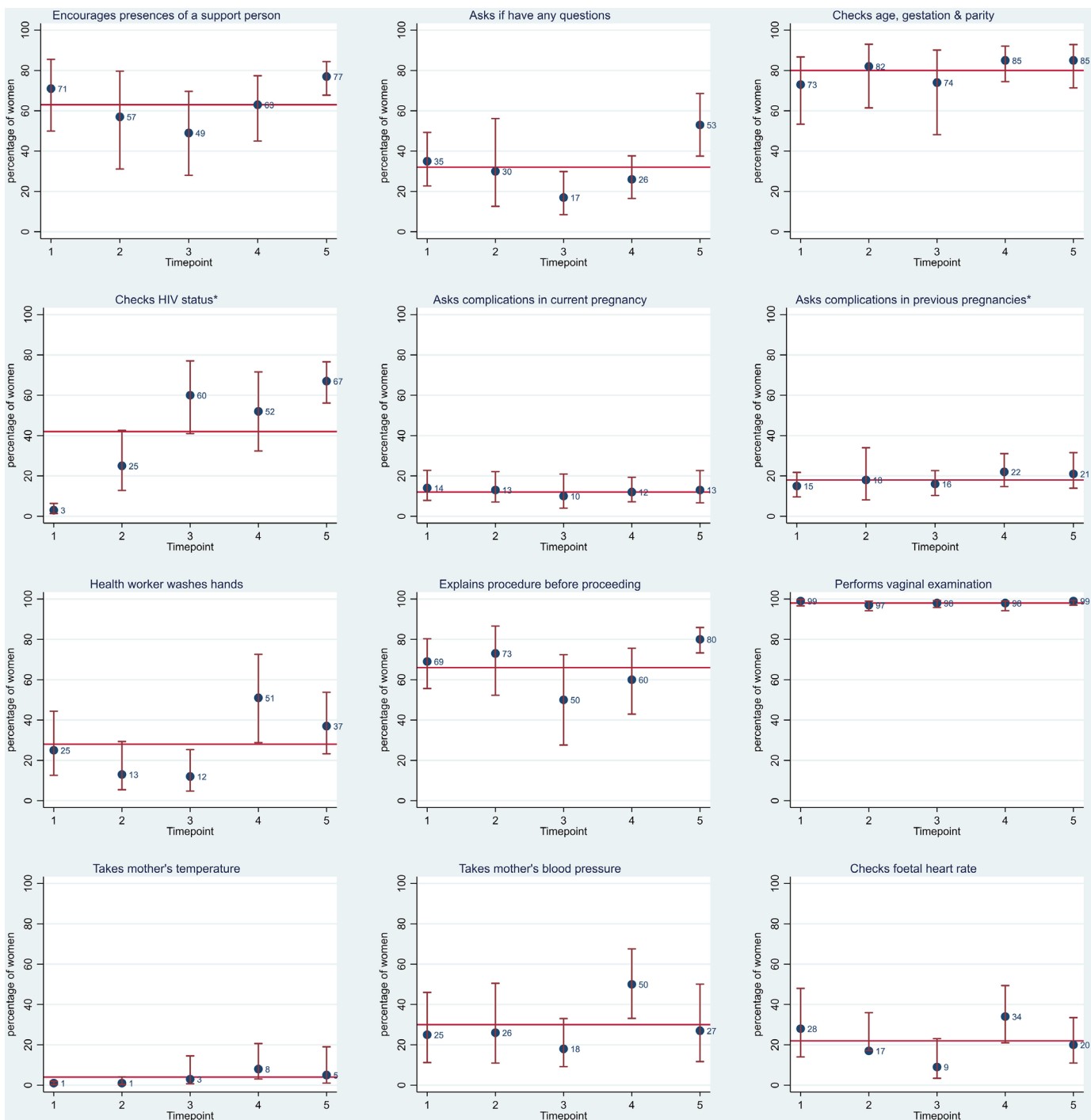

**Figure 2**  History taking and initial examination. Red line represents the average per cent across all five time points. For problems experienced during previous pregnancies, 10% of observers recorded that they did not know if the birth attendant asked the women. For HIV status 30% of observers recorded that they did not know if it had been checked.

if not available, asked women's age, length of pregnancy and parity; 80% (95% CI 73% to 86%), see figure 2. However, other aspects of history taking including asking about complications during both current and, if relevant, previous pregnancies were very low: 12% (95% CI 9% to 16%) and 18% (95% CI 15% to 23%), respectively. The most common complications that birth attendants asked about for the current pregnancy were fever (37%

(81/218)), vaginal bleeding (32% (70/218)) and severe abdominal pain (25% (54/218)).

The majority of women were encouraged to have a support person present during labour and birth (63% (95% CI 53% to 72%)). While around two-thirds of birth attendants explained procedures to woman (and their support person) before proceeding (66% (95% CI 57% to 74%)), less than one-third of women (and their support person)

were asked if they had any questions (32% (95% CI 25% to 40%)).

Vaginal examination was almost universal (98% (95% CI 97% to 99%)) but the proportion of women who had their temperature and blood pressure measured, and the fetal heart rate checked was low: 4% (95% CI 2% to 7%), 30% (95% CI 23% to 39%) and 22% (95% CI 16% to 29%), respectively (figure 2). Few birth attendants washed their hands with soap and water

or antiseptic before examining women (28% (95% CI 21% to 37%)).

## First stage of labour

In total, 1069 women were observed during first stage of labour. Partograph was used to monitor labour in just 24% (95% CI 18% to 32%) of cases (figure 3). Data were not collected on whether there was a delay in the progress of labour but 17% (95% CI 12% to 24%) of women

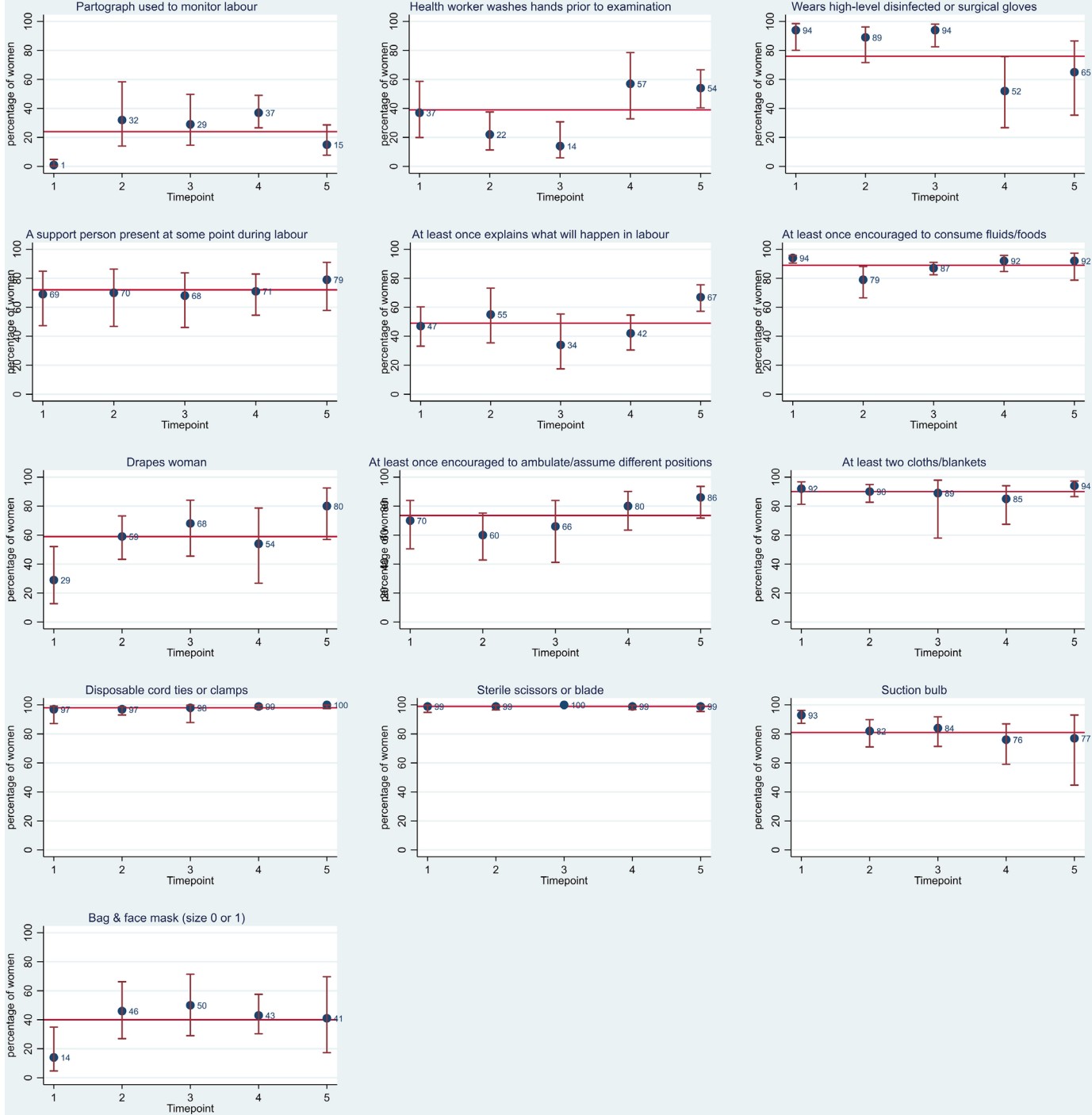

**Figure 3** Examinations and procedures during first stage of labour. Red line represents the average per cent across all five time points.

had their labour augmented with oxytocin and 6% (95% CI 4% to 9%) of women's membranes were artificially ruptured.

The median number of vaginal examinations undertaken during the first stage of labour was one (IQR 1–2). Birth attendants were observed to wash their hands with soap & water or use antiseptic 39% (95% CI 30% to 49%) of the time prior to any examination of women, although a greater proportion were observed to wear high-level disinfectant or surgical gloves when performing the vaginal exam 76% (95% CI 64% to 85%).

Seventy-two per cent (95% CI 63 to 79) of women had a support person present at some point during labour (figure 3). Just under half of birth attendants were observed to explain to the woman (and their support person) what will happen during labour (49% (95% CI 42% to 56%)). The majority of birth attendants were observed to encourage the woman to consume fluids/food at least once during labour (89% (95% CI 85% to 92%)) and to encourage or help woman to ambulate and assume different positions (73% (95% CI 66% to 80%)). During more than half of births observed, birth attendants were observed to drape women with a cloth, one under the buttocks and one over the abdomen (59% (95% CI 48% to 69%)).

Data were collected on whether clean cloths/blankets, tie or cord clamp, sterile blade to cut cord, suction device and bag and mask (either size 0 or 1) were laid out in preparation for birth. Cord clamps, sterile blade and at least two cloths/blankets were available for over 90% of women (figure 3). However, preparation of a bag and mask was substantially lower at 40% (95% CI 31% to 50%).

### Second and third stage of labour

All 1804 women were observed during the second and third stage of labour. Just over half of women observed were assisted at some point during the second and third stage of labour by more than one healthcare worker (57% ((95% CI 50% to 63%)) and 36% (95% CI 27% to 45%) had a support person present (figure 4). Women universally gave birth in the lithotomy position (97% (95% CI 93% to 98%)). The use of episiotomy was very low (1% (95% CI 1% to 2%)). Other inappropriate practices were also low: 3% (95% CI 1% to 7%) fundal pressure, 1% (95% CI 0% to 2%) excessive stretching of perineum and less than 1% had lavage of uterus.

The use of prophylactic uterotonic drugs immediately after birth was universal (96% (95% CI 93% to 98%)), and 65% (95% CI 56% to 73%) of birth attendants checked for the presence of a second baby before administering a uterotonic. Of women that received a uterotonic, 58% received oxytocin (999/1735) and 42% misoprostol (733/1735). Fourteen per cent (95% CI 9% to 20%) of deliveries where uterotonics were administered were given within 1 min of birth and for 52% (95% CI 44% to 57%) of women a uterotonic was administered more than 3 min after birth. Immediately following the delivery/expulsion of the placenta, 75% (95% CI 69% to 80%) of birth attendants were observed to perform uterine massage.

The use of controlled cord traction was consistently undertaken by all cadres of birth attendants: skilled 91% (95% CI 83% to 96%), CHEW 85% (95% CI 76% to 92%) and 'other' birth attendant 84% (95% CI 77% to 89%).

Following the birth of the placenta, 49% (95% CI 39% to 58%) of birth attendants were observed to assess the completeness of placenta and membranes, a considerably higher proportion (87% (95% CI 79% to 92%)) assessed for perineal and vaginal lacerations.

### Immediate newborn and postpartum care

During the immediate postpartum period, 1617 newborns and 1697 women were observed. The majority of newborns received all three elements of thermal care (immediate drying, skin to skin and not bathing in the first hour) and clean cord care (ties or clamps cord when pulsation stop or within 2–3 min after birth and cuts cord with clean scissors or blade): 71% (95% CI 60% to 80%) and 92% (95% CI 89% to 95%), respectively (figure 5). Eighty-six per cent (95% CI 75% to 93%) of newborns had chlorhexidine applied to their cord within the first hour of birth. However, breast feeding was initiated within the first hour after birth in just under half of newborns: 49% (95% CI 39% to 59%).

Eighty-nine per cent (95% CI 79% to 95%) of newborns were weighed (figure 5). Of these infants, 95% (1122/1182) weighed more than 2.5 kg. Mothers and newborns were universally kept in the same room after birth (rooming in), but only 61% (95% CI 51% to 71%) were kept skin to skin during the first hour.

For the fifteen babies born to mothers known to have HIV, 33% (95% CI 14% to 60%) were observed to receive antiretroviral therapy. The proportion of newborns on postnatal care wards who received a vitamin K injection, tetracycline eye ointment and had their temperature checked within 15 min after birth was close to zero throughout the study period (figure 5): 0.1% (95% CI 0% to 0.4%), 4% (95% CI 2% to 7%) and 2% (95% CI 1% to 5%), respectively.

Just 3% (95% CI 2% to 6%) of women had their vital signs checked 15 min after birth and 2% (95% CI 1% to 5%) of women received antibiotics.

### DISCUSSION

Our observations in ten primary healthcare facilities in northeast Nigeria indicated that, while some essential processes of childbirth care were performed for almost all women and newborns, the proportion of women who consistently received evidence-based care during uncomplicated labour was low. Three percent of women (48/1852) who gave birth at the facility experienced major risks and complications while at the facility, including six severe pre-eclampsia/eclampsia and six twin deliveries. Three women died. There were large numbers

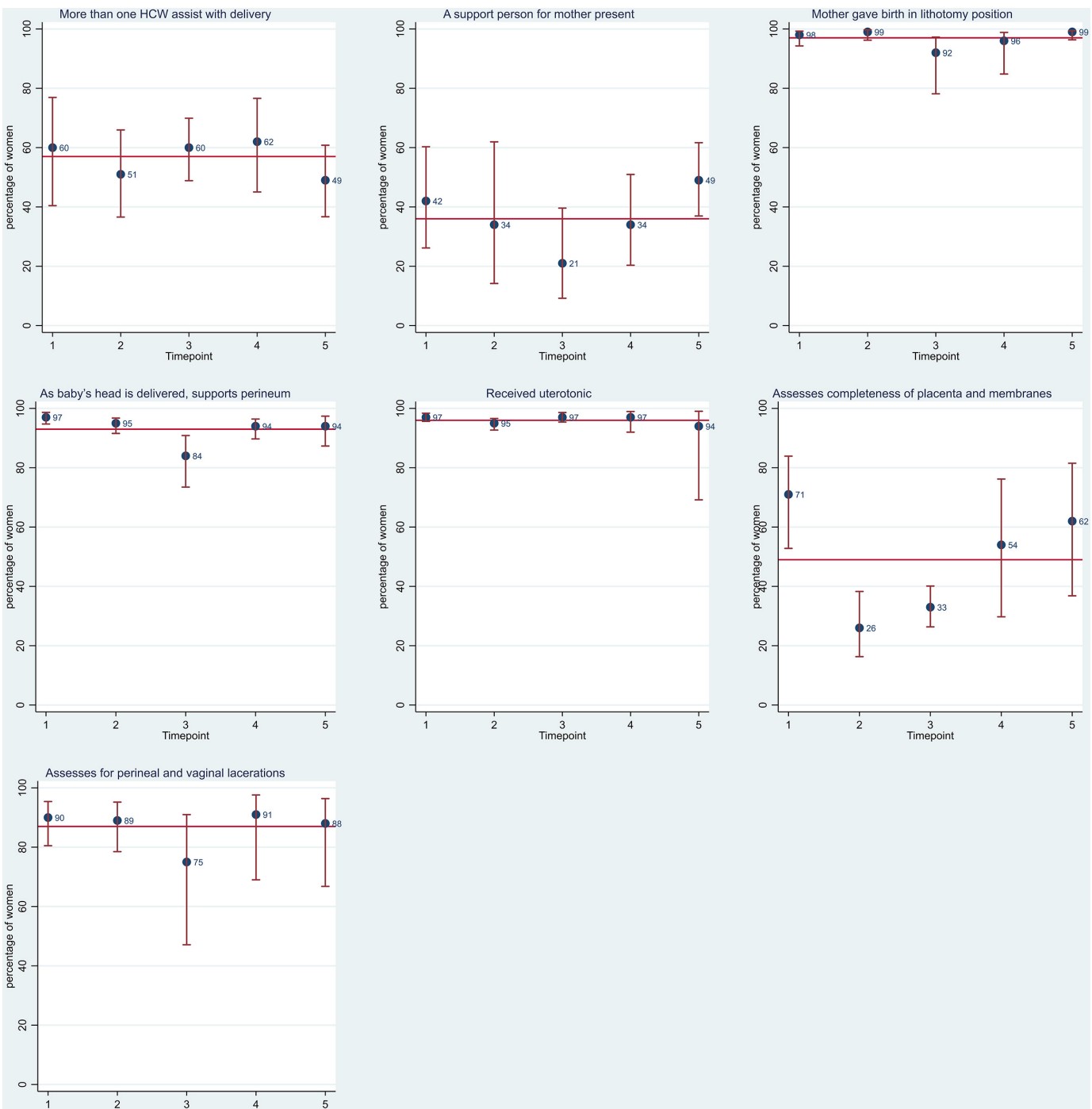

**Figure 4** Care received during second and third stage of labour. Red line represents the average per cent across all five time points. HCW, Healthcare worker.

of perinatal deaths and newborn referrals among this group of women. Further, among women who we defined as having an uncomplicated labour 79 perinatal deaths were recorded, of which almost three-quarters were intrapartum stillbirths.

Our study highlights substantial variation in the implementation of recommended evidence-based interventions both within and across the different stages of childbirth: during all four stages, fewer than half of measures reached more than 80% of women. Implementation was highest

for clinical interventions at the time of birth that have received international attention, for example, the provision of prophylactic uterotonic and newborn thermal and clean cord care. Implementation was lowest for measures designed as risk assessments, for example, history-taking or checking vital signs. All measures related to risk assessment during the initial assessment (asks about complications in current and previous pregnancies, takes temperature and blood pressure, checks fetal heart rate) were completed for fewer than 30% of women observed.

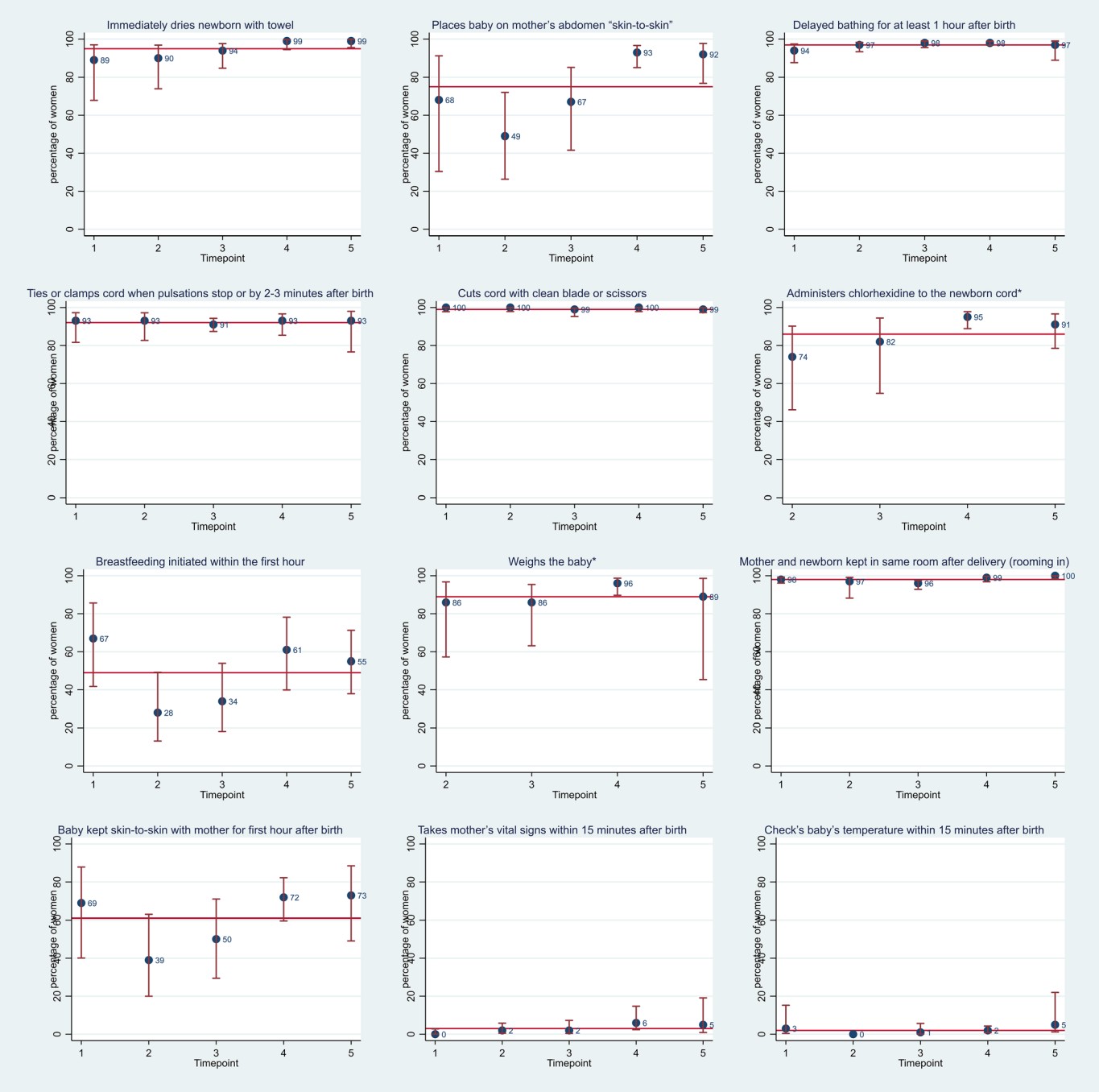

**Figure 5** Immediate newborn and postpartum care. Red line represents the average across all five time points. Administers chlorhexidine and weighs baby was not included in the checklist in the first observation period (time point 1).

The evidence-based care measures included in this study were based on WHO guidelines for essential intrapartum care[15]: their implementation should be covered in basic preservice training for midwives, nurses and CHEWs in this setting; and they require only basic supplies of equipment. However, half of all observed births were attended by 'other' staff who would be defined as unskilled birth attendants both globally and in the Nigerian context: not doctors, nurses or midwives, nor CHEWs who have received additional training under Nigeria's task-shifting policy. Even so, far fewer than half

of the women received many measures, raising questions even about the behaviours of birth attendants who have received training.[43] Further, although some variation was observed over time, in general, coverage of individual measures remained relatively constant throughout the observation period, including for a number of practices that are no longer recommended by WHO, for example, uterine massage.[44] Taken together, this study suggests that there may be limited opportunities for birth attendants to keep their knowledge up to date and enhance their skills once in post. Implementation research is needed

to identify mechanisms to continuously support and improve healthcare worker practices, such as the use of checklists and in-service supervision and coaching, which has shown potential in other settings to improve uptake of and adherence to essential birth practices.[39 45]

A potential alternative reason for the variation in care across the continuum is that some birth attendants were working alone for extended periods. In these circumstances, birth attendants may prioritise interventions during the third stage of labour and immediate newborn care. Higher workload, as a result of staff shortages, has been found to limit healthcare workers time for history taking, thorough assessment of women, and ability to provide timely care.[37 46] Further, when working alone, birth attendants struggle to implement multiple recommended interventions simultaneously, for example, administering uterotonic and providing essential newborn care. While administering uterotonic was universal, less than half of women received a uterotonic within 3 min of birth. The importance of the exact timing of the application of uterotonic is not well established, and as such, it is not clear whether this should be prioritised over immediate newborn care interventions[47 48] and clearer guidelines may be needed if it is important to prioritise certain components of care that cannot be done simultaneously by a single healthcare worker.[49]

Poor assessment of women and newborns before and after birth has previously been documented in India and other African settings.[50–52] Observations of WHO-recommended practices for screening of pre-eclampsia/eclampsia in six sub-Saharan African countries, similarly found that a low proportion of women admitted to labour and delivery services were asked about danger signs, but substantially more women (77%) had their blood pressure checked on admission.[51] Low implementation of evidence-based care measures during the initial assessment and postpartum period indicate potential missed opportunities to identify and manage complications, as evident in the referral cases. Referrals post partum and intrapartum included women with high-risk pregnancies such as severe pre-eclampsia/eclampsia, breech position and twins who should have been referred to a higher level of care.[53] This study has identified proper risk assessment at the time of birth as a priority, not least because, in this setting, only 37% of women attended at least four antenatal care visits and 10% of women and 7% of newborns received a postnatal check within 2 days of birth in 2015.[54]

Poor quality of care has been acknowledged as a critical roadblock in Nigeria's attempt to reach universal health coverage. The Federal government has committed to strengthening Nigeria's health system, particularly primary healthcare, and specifically to accelerate the reduction of maternal and neonatal mortality by expanding access to, and quality of, maternal and child health services.[55] The implementation of the national strategy is supported by Nigeria's participation in the Network for Improving Quality of Care for Maternal, Newborn and Child Health (Quality of Care Network).[56]

The processes of care prioritised in these strategies focus on clinical interventions at the time of birth, including use of uterotonic drugs, skin to skin and chlorhexidine for umbilical cord care—interventions that were found to be routinely well implemented in this low resource primary healthcare setting. This study's findings suggest if these policies are to have an impact they need to extend their focus to also include basic risk-assessment. Further, while they aim to support healthcare workers at the health facilities through quality improvement cycles, clinical mentoring and peer-to-peer learning they may also need to consider the quality of preservice training.

### Strengths and limitations

This study provides unique insight into the provision of evidence-based practices during childbirth and highlights clear areas for action in this setting, actions that are also likely to be relevant elsewhere. A particular strength was the relatively large sample size from multiple time points although, due to the study protocol to prioritise events closest to birth, over 40% of women were not observed during the first stage of labour. The missing observations are not anticipated to impact on findings as non-observation was random, with women who were not observed during the first stage of labour unlikely to differ systematically from those who were. Being observed may have impacted on birth attendants' behaviour; evidence suggests that being observed can positively improve behaviours although any change is likely to be short lived.[57 58] The sample size was estimated to be sufficiently large to reduce the impact of any potential Hawthorne effect and the relative consistency overtime and the low levels of implementation of many measures suggest that any effect of being observed was minimal.

This study was completed in the 10 primary healthcare facilities with the highest volume of births in Gombe state and is not therefore representative of all primary healthcare facilities. It is anticipated that these facilities represent the 'best' care available at the primary level and therefore findings are likely to overestimate the provision of evidence-based care available to the wider population. These findings support the growing body of evidence that giving birth in a primary healthcare facility might not be sufficient to ensure the effective care of women and newborns,[1 4 59] and raise questions about the safety and quality of rural primary healthcare facilities that have low-volume of deliveries.[16 60]

### CONCLUSIONS

The recommendation for women to deliver in health facilities is designed to improve birth outcomes. This study of clinical observations of labour and the immediate postpartum period in primary facilities in Gombe state has revealed that, while some processes of clinical care were well adhered to, most women delivering in primary healthcare facilities do not receive the complete repertoire of childbirth care that they and their newborns

needed to benefit from their choice to deliver in a health facility. In particular, few women or newborns benefited from even basic risk assessments, leading to missed opportunities to identify risks and consequently late referrals and deaths. To continue with the recommendation of childbirth care in primary healthcare in high mortality settings like Gombe it is crucial that birth attendant capacity, capability and prioritisation processes are purposely addressed.

**Acknowledgements** The authors would like to thank Data Research and Mapping Consult for coordinating the data collection, the Gombe State Primary Healthcare Development Agency, Gombe State Ministry of Health and our partners Society for Family Health and Pact Nigeria for their support in carrying out this study. We would like to thank all the participants who contributed to our study.

**Contributors** JE conducted the analysis and drafted the manuscript. CH supported the refinement of our definition of evidence-based care and reviewed the manuscript for intellectual content. NU, BW and AS all critically reviewed the manuscript for intellectual content. TM conceived the study and drafted the manuscript. All authors have read and approved the final version of the manuscript.

**Funding** This work is part of the IDEAS (Informed Decisions for Actions to improve maternal and newborn health) project. IDEAS is funded through a grant from the Bill & Melinda Gates Foundation to the London School of Hygiene & Tropical Medicine. Gates Global Health Grant Number: OPP1149259/INV-007644.

**Disclaimer** The funder had no role in study design, data collection and analysis, decision to publish or preparation of the manuscript.

**Competing interests** None declared.

**Patient consent for publication** Not required.

**Ethics approval** Ethical approval for this study were obtained from the London School of Hygiene & Tropical Medicine (reference 14091) and the Health Research Ethics Committees for Nigeria (reference NHREC/01/01/2007) and Gombe State (reference ADM/S/658/Vol. II/66).

**Provenance and peer review** Not commissioned; externally peer reviewed.

**Data availability statement** Data are available on reasonable request. The data are available from the principal investigator of the IDEAS project and coauthor for this manuscript, TM ORCID id 0000-0002-4228-4334. Reuse permitted on request.

**ORCID iDs**
Josephine Exley http://orcid.org/0000-0002-6501-0854
Tanya Marchant http://orcid.org/0000-0002-4228-4334

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
