## [Reviewer comments · BMJ Open]

ARTICLE DETAILS

TITLE (PROVISIONAL)	The provision of essential evidence-based interventions during facility-based childbirth: cross sectional observations of births in northeast Nigeria
AUTHORS	Exley, Josephine; Hanson, Claudia; Umar, Nasir; Willey, Barbara; Shuaibu, Abdulrahman; Marchant, Tanya

VERSION 1 – REVIEW

REVIEWER	Salisu Ishaku Mohammed Marie Stopes International Nigeria and the Julius center for health science and primary care, University medical center, Utrecht, the Netherlands
REVIEW RETURNED	03-Apr-2020

GENERAL COMMENTS	The article is timely and explored important maternal and child health issues that are relevant for the setting and other similar setting across many LMICs. The authors have taken their time to provide all the details. But in general, I think the paper would benefit from further proofreading to make sure all the punctuation marks and other sentence structures are corrected. Below, I am providing my specific comments on this manuscript • The abstract has no background: part of the requirement for a standard abstract• The objective is not very clear. Are the authors examining level of adherence or compliance to those evidence-based preventive interventions in this setting?• In the abstract, under results section, line 30, the authors provided of estimate of deliveries where uterotonic were used as 90% with a 95CI of 93-98. Ideally the estimate of 90% should in the interval. Why is this not so here. Maybe, the authors could cross check• Study strength and limitation: commencing from line 17, the authors believe that because the facilities included in the study are high-volume sites which may represent where the care are of the highest quality leading to over-estimation of findings. From my experience, the higher the volume the lower the quality as providers are often overburdened. Perhaps the authors may like to reflect on this inference• Under study setting, how were the PHCs chosen out of the remaining? Is it because they are high-volume sites as previously explained or other factors played some roles in their selection? Although this has been briefly mentioned under the data collection process, this should have been indicated under study setting• Under the data collection section, the authors should briefly indicate whether the health care providers were blinded to the purpose of the observers at the facilities. This has implication for bias. For example, if providers were aware that researchers were
---

	there to assess level of compliance with standard of care on the part of their parts, the providers may modify their practices just for that period of observations. This will not reflect the usual care when no one is there. And if the researchers did not take care of risk of bias, the authors should report this as potential weakness in the assessment process.  • The section labelled as 'study population' should have been 'study procedure'. Also, could the authors check whether their definition of uncomplicated labour is evidence-based and provide reference(s)? In my experience, I know 'normal labour' is preferred to 'uncomplicated labor' • The section on consent to participate and ethical approval: The component on informed consent process is better discussed in under the study procedure. The ethical approval should then stand alone • Under the outcomes: Line 54 – 55, the sentence is better described as 'Nine women were admitted with diagnosis of postpartum haemorrhage and 1,866 were admitted with diagnoses of normal labour' • In general, table 1 is always supposed to be show the baseline characteristic of these women. With this, the lengthy description of the women characteristics would have been unnecessary • Results, second and third stage of labour, line 56: This sentence is better placed in the discussion section ('Despite the recent recommendation that uterine massage should not be performed if women have received a uterotonic'). Same on page 13, line 3 • Page 13, line 9: This is phrase 'Following the birth of the placenta' is better replaced with 'Following the delivery of the placenta' • On the discussion: In the first 3 – 4 paragraphs, the authors have repeated many findings already described in the results. The ideal things are for the authors to identify one primary outcome and 2-3 secondary outcomes they would like to discuss further, making sure results are not re-mentioned again. Discussion should be limited to making deductions and opinions from our results as how these results compare to other findings elsewhere as the authors did around paragraph 5. I think everything here but it will be nice if don't mention results again but discussed our finding based on the recommended guidelines and how other studies in this area compare with this • Strength and limitation, line 48: This is better described as" non-observation was likely to be completely at random" not "non-observation was likely to be random"
--	---

REVIEWER	Dr. Aliyah Dosani School of Nursing and Midwifery, Mount Royal University
REVIEW RETURNED	06-Apr-2020

GENERAL COMMENTS	Thank you for submitting this article. It's content is interesting and most important to ensure the health of mothers and infants worldwide. I provide my comments below:  1) In your abstract, in the objectives section you indicate "a high-mortality setting". Please clarify if you are referring to expectant mothers, infants or both. 2) In your abstract, there is a bit of disconnect between your results section and your conclusion. In your conclusion, section you indicate that the majority of women did not receive the recommended routine processes of childbirth care. Are you referring to the composite score of 10 essential items here? Please clarify. 3) Page 5 of 35, line 23 - you state "recommended interventions are practiced...". Please clarify whomb your research is focussed on
---

here? Is it trained medical or nursing professionals, or skilled birth attendants. Can you please focus your literature review to which cadre of health worker or professionals you are focussing your 10 essential itmes?

4) Page 5 of 35, the last paragraph belongs in the methods section.

5) Please include the objective/research questions as the last paragraph in your introduction.

5) Page 5 of 35, Methods section, line 46, you state: "We conducted direct observations of childbirth care in 10 primary healthcare facilities". Please clarify what cadre of worker was providing childbirth care.

6) Informaiton you provide in your "study setting" section of methods may be better suited infromation to include in your introduction. Please revise accordingly.

7) Please provide your rationale for using a cross-sectional data collection method.

8) Page 8 of 35 line 31 - I think you mean inclusion criteria here. Please carefully use terminology in appropriate sections. A discussion of your study population belongs in your introduction.

9) Page 9 of 35, results section. You state: "The median gestational age was 39 weeks (IQR 38–39); 6% of women had a gestational age of less than 37 weeks and for 21% of women gestational age was not recorded on their client card and/or they did not know." Are you referring to gestational age age birth? Please clarify. Also, can you please provide more information around why you chose to present the median gestational age of 39 weeks, rather than the mean? I may make more sense to present the mean.

10) In the results section, you state " Eleven percent of women were seen by a skilled birth attendant (doctor, midwife, nurse), 39% by a community health extension worker (CHEW) or junior CHEW and 50% by an 'other' birth attendant. 'Other' included traditional birth attendants, environmental health officers/technicians/assistants and health attendant/assistant." Are you able to identify at which state along the continuum of pregnancy these types of health workers were seen and at what point (if any) did the cadre of health worker change?

11) page 10 of 35 - please double check your terminology with respect to "panel 1". Is this terminology consistent with author guidelines for submission?

12) Page 11 of 35, lines 35 and 42. Please check your formatting.

13) In your methods section, please discuss how your arrived at the 10 criteria used for the composite score. What made some criteria more desireable to capture than others? Please provide your rationale.

14) It may make sense to provide an abridged version of your supplementary table that corresponds to the 10 item composite score in your results section. It would also make sense for you to present the results in written form as they relate to the 10 item composite score.

15) Page 14 of 35 - you speak to guidelines for midwives, nurses, and community health workers. Please ensure that you are discussing your results based on the cadre of health care worker that you have observed as part of your study.

16) After the results have been ammended, changes may need to be made to the discussion based on the reorganization of your results.

17) While Figures 1-5 are useful, it may make more sense to present this information in table format so that one may be able to see the trends over time with respect to each component.

Thank you for preparing a thorough manuscript.

REVIEWER	Corrina Moucheraud University of California Los Angeles, USA
REVIEW RETURNED	10-Apr-2020

GENERAL COMMENTS	This was a very well-written and interesting manuscript, on an important topic. I think several changes to the presentation of methods and results could help increase the significance of this paper.  1. I would suggest including more information about how the measures were decided upon and how they correspond to the other checklists (maybe an appendix table that crosswalks each item to what other checklist(s) it appears on). 2. Why were the bolded items included in the composite score but not others? I feel the authors need to provide justification here. Additionally, I noticed that nothing from postpartum care was included in the composite score -- why not? 3. Since priority was given to observing the second and third stages of labor, how did authors handle missingness for items (especially during the earlier labor stages that were less likely to be observed)? This is particularly salient for the composite measure - were women without full datasets excluded from this analysis (which, based on Table 1, would reduce the sample size for this analysis to only 1069 women)? 4. I don't see information in Methods about how the women's characteristics were collected. Suggest including this. 5. Given how common "other" birth attendants were (50% of the observed births), I suggest disaggregating this category. 6. For the 17 babies who died after attempted resuscitation: are they included in the 96 who received resuscitation? Or were those 96 all successful resuscitations? 7. Panel 1 does not seem very informative to me; these #s should be reported in the results (tables etc.) if they are relevant to the findings of this study -- and the quotes do not really add new or particularly illuminating information, in my opinion. 8. I feel the results would be more compelling (& actionable by programs and policies) if the authors showed how these %s differed by provider type, or by the woman's characteristics. 9. It is unclear to me why the figures (& supplementary material) stratify by time point. Wouldn't a pooled analysis give more power to detect differences? Is there a hypothesis about changes over time? If so this should be clearer earlier in the article, when the research aims are presented etc. 10. Additionally, the statistics reported in-text are presumably pooled across time periods. The mismatch in reporting between the text & the figures is challenging for the reader to get a clear picture of the results. 11. One possible use of the time-varying results is to test for potential Hawthorne effects. This would be informative as a
---

	sensitivity analysis (perhaps omitting the early observations & assessing if the relationships still hold). I would also suggest mentioning observation bias in the Limitations section.
--	--

REVIEWER	Dr. Alex Ernest The University of Dodoma, College of Health Sciences, Tanzania
REVIEW RETURNED	13-Apr-2020

GENERAL COMMENTS	 ● The study is of public importance and addressed important issues especially in low income communities hence worth publication. ● The title reflect the objectives and well stated. ● The introduction capture the main theme of the study. There are some few typos error ,example, the last sentence of the last paragraph, the newborn instead of their newborn ● The methodology need some improvements and raise eyebrow on some areas. For instance, the observations was done between six months in a 2 year period. The interval time in between could be affected by on going interventions by NGO's or even continuous medical education. This should be clearly addressed. ● The so called high volume facilities could not necessary reflect the quality of care especially in settings of low socioeconomic. This should be interpreted with caution especially in low income countries. Given the circumstances, in most instances, the high volume facilities succumb to poor quality of services which might've partially explain your findings. A random inclusion of low volume facilities could have given the true burden of the problem and a pertinent conclusion. ● There is a major issue of ethical consideration in this study. This was unobstructed observation study. The study didnt explain clearly on how they handle cases where the standards were not adhered as it could have lead to adverse consequences. Your study recorded 3 maternal death and about 79 newborn death which were basically preventable. What if someone attribute it to nonobstructive nature of the study? This need to be clearly explained in this study. At some point you mentioned that the observer prioritize the safety of mothers and newborns. This sound contradictory to your unobstructed observation. Did you exclude cases where observers due to some reasons interfere with management to save the life of mother or newborn? This again need explanation. Looking on observers free comments it suggest that your study was completely unobstructive and hence raise questions on ethical issues. It is clear that observation study poses risk of bias and may affect the practice positively. Did you include this as the limitation of your study and how did you overcome it. This need to be adressed as well. ● Your results could have been affected by many confounding factors including facility budget, staffing level, availability of working tools like thermometer, BP machines, partography , available interventions, etc etc. How did you take into account of all these confounding factors. Was all the factors constant in all these health facilities? ● The discussion is well written and reflect the results findings. There are some few typos like in one sentence it is written, however haf instead of however half. ● The conclusion of the study cannot be generalised to the population. The disparity between facilities should've been clearly shown and each addressed separately.
---

VERSION 1 – AUTHOR RESPONSE

Reviewer comments

Comment	Reviewer	
	1	Salisu Ishaku Mohammed, Marie Stopes International Nigeria and the Julius center for health science and primary care, University medical center, Utrecht, the Netherlands. The article is timely and explored important maternal and child health issues that are relevant for the setting and other similar setting across many LMICs. The authors have taken their time to provide all the details. But in general, I think the paper would benefit from further proofreading to make sure all the punctuation marks and other sentence structures are corrected. Below, I am proving my specific comments on this manuscript
	2	Dr. Aliyah Dosani, School of Nursing and Midwifery, Mount Royal University. Thank you for submitting this article. It's content is interesting and most important to ensure the health of mothers and infants worldwide. I provide my comments below:
	3	Corrina Moucheraud, University of California Los Angeles, USA. This was a very well-written and interesting manuscript, on an important topic. I think several changes to the presentation of methods and results could help increase the significance of this paper.
	4	Dr. Alex Ernest, The University of Dodoma, College of Health Sciences, Tanzania. The study is of public importance and addressed important issues especially in low income communities hence worth publication.
Abstract		
1	1	The abstract has no background: part of the requirement for a standard abstract We acknowledge that the abstract is not written in a traditional format, however it is presented in line with the structure requested by the journal.
2	1	The objective is not very clear. Are the authors examining level of adherence or compliance to those evidence-based preventive interventions in this setting? The wording has now been updated as follows: "To measure the provision of evidence-based preventive and promotive interventions to women, and subsequently their newborns, during childbirth in a high-mortality setting." The aim was not to examine adherence or compliance but to provide a comprehensive picture of which components of care are routinely being provided in this setting or missed.
3	2	In the objectives section you indicate "a high-mortality setting". Please clarify if you are referring to expectant mothers, infants or both. We have now provided additional details in the setting to indicate that this refers to both mothers and newborns: “Setting: Ten primary healthcare facilities in Gombe State, northeast Nigeria. The northeast region of Nigeria has some of the highest maternal and newborn death rates globally.”
4	1	In the abstract, under results section, line 30, the authors provided of estimate of deliveries where uterotonic were used as 90% with a 95CI of 93-98. Ideally the estimate of 90% should in Thank you for picking this up – this is indeed an error and should be 96% [95%CI 93-98] as presented in the results section. This has now been updated.

Comment	Reviewer		
		the interval. Why is this not so here. Maybe, the authors could cross check	
5	1	Study strength and limitation: commencing from line 17, the authors believe that because the facilities included in the study are high-volume sites which may represent where the care are of the highest quality leading to over-estimation of findings. From my experience, the higher the volume the lower the quality as providers are often overburdened. Perhaps the authors may like to reflect on this inference	We agree with the point being raised by the reviewer and in the discussion have reflected on the impact that a higher workload might have on birth attendants' ability to deliver high quality care, and suggest this might be a reason for limited history taking and thorough assessment of the women. We have now updated the bullet to remove text that suggests high volume represents the best quality. It now reads as follows: "This study was completed in the 10 primary care facilities with the highest volume of births in Gombe State and therefore are not representative of all facilities." In response to this and other comments, we have reworded throughout to make it clear that these are not 'high volume' facilities. These are the 10 facilities with the highest volume of births. Work by Kruk et al. (reference 16 in the manuscript) found very low quality of delivery care in facilities with low delivery volume of less than 500 births per year (41.7 per month). In the year before data collection commenced the mean number of deliveries per month in the 10 PHCs where observations took place was 15.7 (sd 12.0). Across the state as a whole the mean number of births per month was 4.3 (sd 6.3) births per facility per month.
6	4	The so called high volume facilities could not necessary reflect the quality of care especially in settings of low socioeconomic. This should be interpreted with caution especially in low income countries. Given the circumstances, in most instances, the high volume facilities succumb to poor quality of services which might've partially explain your findings. A random inclusion of low volume facilities could have given the true burden of the problem and a petinent conclusion.	Please see response to comment 5, the volume of facility births in this setting is low. To achieve a sufficiently large sample size we used the volume of deliveries to inform facility sample selection. These facilities are therefore not likely to be representative.
7	2	There is a bit of disconnect between your results section and your conclusion. In your conclusion, section you indicate that the majority of women did not receive the recommended routine processes of childbirth care. Are you referring to the composite score of	Thank you for this observation. This along with other reviewers' comments has made us reflect on the utility of the composite score and we have made the decision to remove this from the manuscript.

Comment	Reviewer		
		10 essential items here? Please clarify.	
Introduction			
8	2	Page 5 of 35, line 23 - you state "recommended interventions are practiced...". Please clarify whom your research is focussed on here? Is it trained medical or nursing professionals, or skilled birth attendants. Can you please focus your literature review to which cadre of health worker or professionals you are focussing your 10 essential items?	We have now added additional details to the study setting to highlight that this research was conducted in the context of a human resource crisis. In Gombe State, skilled healthcare providers including medical doctors and nurses/midwives constitute only 4% and 27% of the total health workforce, respectively. Doctors predominantly work in tertiary hospitals and are therefore absent from PHCs. The research was not focused on a particular cadre. The intention was to provide a snap shot of the 'usual care' that women who attend for facility based delivery are likely to receive. This includes the finding that the majority of women are not attended by a skilled health care providers in this setting.
9	2	Page 5 of 35, the last paragraph belongs in the methods section	We have now edited the wording of the final paragraph to make it more explicit that this was the aim of the study. It now reads as follows:
10	2	Please include the objective/research questions as the last paragraph in your introduction	"In this study of birth observations, we aimed to examine the maternal and newborn outcomes experienced by all women admitted for childbirth and postpartum haemorrhage in a sample of primary healthcare facilities in Gombe State, Nigeria. For women who had an uncomplicated labour, we evaluated the provision of evidence-based care provided to women, and subsequently their newborns, from initial assessment up to one hour postpartum."
11	4	The introduction capture the main theme of the study. There are some few typos error ,example, the last sentence of the last paragraph, theinewborn instead of their newborn	Thank you for your careful review.
Methods			
12	2	Page 5 of 35, Methods section, line 46, you state: "We conducted direct observations of childbirth care in 10 primary healthcare facilities". Please clarify what cadre of worker was providing childbirth care.	This was not restricted to any particular cadre and we have now clarified this in the data collect section of the methods: "Observers aimed to observe all women who were admitted irrespective of the cadre of the attending healthcare worker" In the opening section of the results (page 9 line 187 onwards) we highlight who the birth attendant was. See also response to comment 8, we have now included additional information in the study setting to

Comment	Reviewer		
			highlight in this setting this is predominantly lower cadres of healthcare workers.
13	1	Under study setting, how were the PHCs chosen out of the remaining? Is it because they are high-volume sites as previously explained or other factors played some roles in their selection? Although this has been briefly mentioned under the data collection process, this should have been indicated under study setting	We have separated the information on sample selection from the data collection section to improve readability.
14	2	Information you provide in your "study setting" section of methods may be better suited information to include in your introduction. Please revise accordingly.	Thank you for your comment. We have restructured various sections of the paper following review, and believe that it now provides an appropriate and comprehensive report of our study.
15	2	Please provide your rationale for using a cross-sectional data collection method.	This study was designed to provide a detailed snap shot of quality of care in Gombe state.
16	1	Under the data collection section, the authors should briefly indicate whether the health care providers were blinded to the purpose of the observers at the facilities. This has implication for bias. For example, if providers were aware that researchers were there to assess level of compliance with standard of care on the part of their parts, the providers may modify their practices just for that period of observations. This will not reflect the usual care when no one is there. And if the researchers did not take care of risk of bias, the authors should report this as potential weakness in the assessment process.	We agree with the reviewer that there is a risk of bias as a result of being observed. We undertook direct observations and as such birth attendants were not blinded. Every effort was made to ensure that observations were undertaken unobtrusively and to put the birth attendant and women at ease. We have added additional information on role of the clinical supervisor in the data collection methods. In anticipation of any potential Hawthorne effect, the sample size estimates were calculated to be sufficiently large to account for the impact. Evidence has shown that the impact is to improve behaviours, but that any change is short lived. Sustained observations over a period of time has been found to mitigate against some of the impacts. We have added additional detail and references to the limitation section of the discussion.
17	3	I don't see information in Methods about how the women's characteristics were collected. Suggest including this.	Women's characteristics were collected as part of the observation checklist; details have been added to the data collection section.
18	3	I would suggest including more information about how the measures were decided upon and how they correspond to the other checklists (maybe an appendix table that crosswalks each item to what other checklist(s) it appears on).	We are attaching the mapping exercise along with our response. We would be happy to include as supplementary material if the editors approve.
19	1	The section labelled as 'study population' should have been 'study procedure'.	This section has now been relabelled inclusion criteria.

Comment	Reviewer		
20	2	Page 8 of 35 line 31 - I think you mean inclusion criteria here. Please carefully use terminology in appropriate sections. A discussion of your study population belongs in your introduction	
21	1	Also, could the authors check whether their definition of uncomplicated labour is evidence-based and provide reference(s)?. In my experience, I know 'normal labour' is preferred to 'uncomplicated labor'	After consultation with co-authors, we respectfully prefer to keep as uncomplicated.
22	2	In your methods section, please discuss how you arrived at the 10 criteria used for the composite score. What made some criteria more desirable to capture than others? Please provide your rationale.	Please see response to comment 7
23	3	Why were the bolded items included in the composite score but not others? I feel the authors need to provide justification here. Additionally, I noticed that nothing from postpartum care was included in the composite score -- why not?	
24	3	Since priority was given to observing the second and third stages of labor, how did authors handle missingness for items (especially during the earlier labor stages that were less likely to be observed)? This is particularly salient for the composite measure - were women without full datasets excluded from this analysis (which, based on Table 1, would reduce the sample size for this analysis to only 1069 women)?	The denominator for each stage is the number of women observed during that stage (presented in Table 1). In general data completeness was high; we indicate where missing data was greater than 10% in the figure legends (this was the case for only 2 measures: problems experienced during previous pregnancies & checking HIV status). In the limitations section we discuss the potential impact of non-observation.
25	3	It is unclear to me why the figures (& supplementary material) stratify by time point. Wouldn't a pooled analysis give more power to detect differences? Is there a hypothesis about changes over time? If so this should be clearer earlier in the article, when the research aims are presented etc Additionally, the statistics reported in-text are presumably pooled across time periods. The mismatch in reporting between the text & the figures is challenging for the reader to get a clear picture of the results.	We agree with the reviewers that given the study was conducted over a two-year time period, we might have anticipated changes overtime. To provide transparency we therefore present the results broken down by time point. However, it can be seen that whilst there is some variability there were no time trends, so we present the pooled results in the text. We have clarified in the analysis section of the methods that in the text we present the pooled results.
26	4	The methodology needs some improvements and raise eyebrows on some areas. For instance, the observations were done between six months in a 2-year period. The interval time in between could be affected by ongoing interventions by NGOs or even	

Comment	Reviewer		
		continuous medical education. This should be clearly addressed.	
28	3	One possible use of the time-varying results is to test for potential Hawthorne effects. This would be informative as a sensitivity analysis (perhaps omitting the early observations & assessing if the relationships still hold).	Our understanding is that the hawthorne effect would occur within time periods rather than between data collection periods. There would have been other differences between time periods such as the staff observed and the individual conducting the observations.
27	4	Your results could have been affected by many confounding factors including facility budget, staffing level, availability of working tools like thermometer, BP machines, partography , available interventions, etc etc. How did you take into account of all these confounding factors. Was all the factors constant in all these health facilities?	We present a univariate analysis, and therefore did not control for confounders. It is likely that there were difference between facilities and overtime. We have allowed for clustering by facility and changes over time in the analysis through the use of the svyset function in stata to take account of these survey features and resent the 95%CI. .
28	1	The section on consent to participate and ethical approval: The component on informed consent process is better discussed in under the study procedure. The ethical approval should then stand alone	We have now moved the details on consent procedure under data collection.
29	4	There is a major issue of ethical consideration in this study. This was unobstructed observation study. The study didnt explain clearly on how they handle cases where the standards were not adhered as it could have lead to adverse consequences. Your study recorded 3 maternal death and about 79 newborn death which were basically preventable. What if someone attribute it to nonobstuctive nature of the study? This need to be clearly explained in this study. At some point you mentioned that the observer prioritize the safety of mothers and newborns. This sound contradictory to your unobstructed observation. Did you exclude cases where observers due to some reasons interfere with management to save the life of mother or newborn? This again need explanation.Looking on observers free comments it suggest that your study was completely unobstructive and hence raise questions on ethical issues.	We acknowledge that there are serious ethical implications of clinical observations. The study was thoroughly reviewed by authorities at state and national level in Nigeria. The safeguarding mechanisms put in place were recommended and approved by national and state level government. The first protocol for observers to follow is to call for help if they judge that the client is in danger. In reality, many health facilities in this setting have only one trained health worker available to provide care and so additional clinical help may not be routinely available. This is one reason why the observation work was closely supervised by a clinician. After consultation with the State government, however, we were constrained by the restriction in law that a health worker may not legally practise in a government facility without prior registration. So the supervisor could advise but not intervene. In cases where observers intervned data collection was stopped and these women were excluded from the study.

Comment	Reviewer		
			There is a precedent for this type of work. See for example: Day, L. T., et al. (2019). "Every Newborn-BIRTH" protocol: observational study validating indicators for coverage and quality of maternal and newborn health care in Bangladesh, Nepal and Tanzania." J Glob Health 9(1): 010902
Results			
30	1	Under the outcomes: Line 54 – 55, the sentence is better described as ‘Nine women were admitted with diagnosis of postpartum haemorrhage and 1,866 were admitted with diagnoses of normal labour’	We are not able to determine if the 1,866 women were admitted with a diagnosis of normal labour. What we observed was a number of these women were referred out during labour and then a number experienced a complication post-delivery.
31	1	In general, table 1 is always supposed to be show the baseline characteristic of these women. With this, the lengthy description of the women characteristics would have been unnecessary	We would be happy to include a table of women’s characteristics if the editor approves.
32	2	Page 9 of 35, results section. You state: "The median gestational age was 39 weeks (IQR 38–39); 6% of women had a gestational age of less than 37 weeks and for 21% of women gestational age was not recorded on their client card and/or they did not know." Are you referring to gestational age at birth? Please clarify. Also, can you please provide more information around why you chose to present the median gestational age of 39 weeks, rather than the mean? I may make more sense to present the mean.	We have revised the text to make it clear that this was the age recorded at admission: “The median gestational age of women on admission was 39 weeks (IQR 38–39)” The median was selected as it better represents the distribution of the data than the mean.
33	2	In the results section, you state "Eleven percent of women were seen by a skilled birth attendant (doctor, midwife, nurse), 39% by a community health extension worker (CHEW) or junior CHEW and 50% by an ‘other’ birth attendant. ‘Other’ included traditional birth attendants, environmental health officers/technicians/assistants and health attendant/assistant." Are you able to identify at which state along the continuum of pregnancy these types of health workers were seen and at what point (if any) did the cadre of health worker change?	The study only looked at facility delivery and so we cannot comment on the state along the continuum of pregnancy. The cadre was recorded at the start of the observation period. We are not able to determine if it changed.
34	3	Given how common "other" birth attendants were (50% of the observed births), I suggest disaggregating this category.	We have now added the percentage for each of ‘other’ birth attendants. “‘Other’ included environmental health officers/technicians/assistants (43%), health attendant/assistant (43%), traditional birth attendants (4%),

Comment	Reviewer		
			community health officer (1%) and other (9%)."
35	3	I feel the results would be more compelling (& actionable by programs and policies) if the authors showed how these %s differed by provider type, or by the woman's characteristics.	We thank the reviewer for this comment. The study was not powered to examine difference between cadres, however we have looked at the data by the different cadres and observed no consistent patterns. We see clear gaps in the provision of care, especially related to risk assessment which are not being completed by any cadre of health worker. We feel that the analysis is presented in such a way that reflects the actual experience of women attending facilities for childbirth care in this setting.
36	3	For the 17 babies who died after attempted resuscitation: are they included in the 96 who received resuscitation? Or were those 96 all successful resuscitations?	We have now updated the text to make this clearer that this includes the babies that died following resuscitation: "For the babies born to women who had an uncomplicated labour, 96 received resuscitation care of whom 19% subsequently died. Overall, 79 babies died: the perinatal mortality rate was 44.1 per 1,000 newborns"
37	2	page 10 of 35 - please double check your terminology with respect to "panel 1". Is this terminology consistent with author guidelines for submission?	Could the editors please advise as we cannot find any guidance on this in the instructions for authors.
38	3	Panel 1 does not seem very informative to me; these #s should be reported in the results (tables etc.) if they are relevant to the findings of this study -- and the quotes do not really add new or particularly illuminating information, in my opinion.	The data is relatively unique and therefore think that it is worth keeping, but are happy to refer to the editor whether this should be included. The numbers are presented in figure 1 and they are included in the opening description of the characteristics of women that were admitted to the PHCs. For the main analysis these women are excluded as we do not classify them as having an uncomplicated labour.
39	2	Page 11 of 35, lines 35 and 42. Please check your formatting	Thank you for your careful review.
40	1	second and third stage of labour, line 56: This sentence is better placed in the discussion section ('Despite the recent recommendation that uterine massage should not be performed if women have received a uterotonic'). Same on page 13, line 3	Both sentences have been removed from the results section.
41	2	It may make sense to provide an abridged version of your supplementary table that corresponds to the 10 item composite score in your results section. It would also make sense for you to present the results in written form as they relate to the 10 item composite	We have now removed the composite score.

Comment	Reviewer		
		score.	
42	2	While Figures 1-5 are useful, it may make more sense to present this information in table format so that one may be able to see the trends over time with respect to each component.	A full table of results was attached as supplementary material.
Discussion			
43	1	Page 13, line 9: This is phrase 'Following the birth of the placenta' is better replaced with 'Following the delivery of the placenta'	
44	1	In the first 3 – 4 paragraphs, the authors have repeated many findings already described in the results. The ideal things are for the authors to identify one primary outcome and 2-3 secondary outcomes they would like to discuss further, making sure results are not re-mentioned again. Discussion should be limited to making deductions and opinions from our results as how these results compare to other findings elsewhere as the authors did around paragraph 5. I think everything here but it will be nice if don't mention results again but discussed our finding based on the recommended guidelines and how other studies in this area compare with this	Thank you for the comment – we have made edits throughout the manuscript to improve readability.
45	2	Page 14 of 35 - you speak to guidelines for midwives, nurses, and community health workers. Please ensure that you are discussing your results based on the cadre of health care worker that you have observed as part of your study.	The majority of health care workers observed would not be classified as skilled by international definition. We provide a more thorough explanation of the context, and also on page 14 line 352, we discuss the fact that these health care workers are excluded from training programme.
46	2	After the results have been ammended, changes may need to be made to the discussion based on the reorganization of your results.	No further action required.
47	4	The discussion is well written and reflect the results findings. There are some few typos like in one sentence it is written, however haf instead of however half.	We thank the reviewer for their careful review.
Strengths and limitations			
48	1	, line 48: This is better described as" non-observation was likely to be completely at random" not "non-observation was likely to be random"	This has been updated.
49	3	I would also suggest mentioning observation bias in the Limitations section.	Please see response to comment 16
50	4	It is clear that observation study poses risk of bias and may affect the practice positively. Did you include this as the limitation of your study and how did you	

Comment	Reviewer	
		overcome it. This need to be adressed as well.
Conclusions		
51	4	The conclusion of the study cannot be generalised to the population. The disparity between facilities should've been clearly shown and each addressed separately.
		We acknowledge that these findings are not generalisable and have discussed this on page 15 line 422 onwards.

VERSION 2 – REVIEW

REVIEWER	Salisu Ishaku Mohammed Julius center for health and primary care, university medical center Utrecht, the Netherlands and Marie Stopes International Nigeria
REVIEW RETURNED	20-Jul-2020

GENERAL COMMENTS	Although the authors did not include a background to the abstract as suggested, the study setting and objective provided the required background to the study. The objective is clearer than of the original version. The initial error in the confidence interval around the 96% provision of uterotonics has been corrected. The concern I earlier expressed in the strength and limitation section of the abstract where the authors linked high-volume facilities with improve quality of care has also been corrected. With this, the abstract looks perfect from my perspective. The authors have clarified what the impact of being observed could have on the quality of care on the health care workers that could lead to bias. The original paper has labelled inclusion criteria as study population. This version has corrected the error. The mixing together of consent process and the ethical approval in the original draft have also been presented separately. The discussion is much improved. In general, I believe that the article in this revised version is publishable in the BMJ Open
---

REVIEWER	Corrina Moucheraud University of California Los Angeles, USA
REVIEW RETURNED	02-Jul-2020

GENERAL COMMENTS	I feel that the authors have comprehensively and thoughtfully adressed my and the other reviewers' comments, and the manuscript is strengthened as a result. Congratulations on an important and well-written article.
--

REVIEWER	Alex Ernest The University of Dodoma
REVIEW RETURNED	05-Aug-2020

GENERAL COMMENTS	The authors have addressed most of the issues raised.
---

VERSION 2 – AUTHOR RESPONSE

Response to reviewer 2's comments submitted 29th June 2020:

Comment by reviewer 2	Response
1 In the objectives section you indicate "a high-mortality setting". Please clarify if you are referring to expectant mothers, infants or both.	We have now provided additional details in the setting to indicate that this refers to both mothers and newborns: “Setting: Ten primary healthcare facilities in Gombe State, northeast Nigeria. The northeast region of Nigeria has some of the highest maternal and newborn death rates globally.”
2 There is a bit of disconnect between your results section and your conclusion. In your conclusion, section you indicate that the majority of women did not receive the recommended routine processes of childbirth care. Are you referring to the composite score of 10 essential items here? Please clarify.	Thank you for this observation. This along with other reviewers' comments has made us reflect on the utility of the composite score and we have made the decision to remove this from the manuscript.
3 Page 5 of 35, line 23 - you state "recommended interventions are practiced...". Please clarify whom your research is focussed on here? Is it trained medical or nursing professionals, or skilled birth attendants. Can you please focus your literature review to which cadre of health worker or professionals you are focussing your 10 essential items?	We have now added additional details to the study setting section of the methods to highlight that this research was conducted in the context of a human resource crisis. In Gombe State, skilled healthcare providers including medical doctors and nurses/midwives constitute only 4% and 27% of the total health workforce, respectively. Doctors predominantly work in tertiary hospitals and are therefore absent from PHCs. The research was not focused on a particular cadre. The intention was to provide a snap shot of the 'usual care' that women who attend for facility based delivery are likely to receive. This includes the finding that the majority of women are not attended by a skilled health care providers in this setting.
4 Page 5 of 35, the last paragraph belongs in the methods section	We have now edited the wording of the final paragraph to make it more explicit that this was the aim of the study. It now reads as follows:
5 Please include the objective/research questions as the last paragraph in your introduction	“In this study of birth observations, we aimed to examine the maternal and newborn outcomes experienced by all women admitted for childbirth and postpartum haemorrhage in a sample of primary healthcare facilities in Gombe State, Nigeria. For women who had an uncomplicated labour, we evaluated the provision of evidence-based care provided to women, and subsequently their newborns, from initial assessment up to one hour postpartum.”
5 Page 5 of 35, Methods section, line 46, you state: "We conducted direct observations of childbirth care in 10 primary healthcare facilities". Please clarify what cadre of worker was providing childbirth care.	We have now clarified this in the data collect section of the methods: “Observers aimed to observe all women who were admitted irrespective of the cadre of the attending healthcare worker” In the opening section of the results (page 9 line 187 onwards) we highlight who the birth

Comment by reviewer 2	Response
	attendant was. See also response to comment 3, we have now included additional information in the study setting to highlight that, in this setting, care is predominantly provided by lower cadres of healthcare workers.
6 Information you provide in your "study setting" section of methods may be better suited information to include in your introduction. Please revise accordingly.	Thank you for your comment. We have restructured various sections of the paper following review, and believe that it now provides an appropriate and comprehensive report of our study.
7 Please provide your rationale for using a cross-sectional data collection method.	This study was designed to provide a detailed snap shot of quality of care in Gombe state.
8 Page 8 of 35 line 31 - I think you mean inclusion criteria here. Please carefully use terminology in appropriate sections. A discussion of your study population belongs in your introduction	This section has now been relabelled inclusion criteria.
9 Page 9 of 35, results section. You state: "The median gestational age was 39 weeks (IQR 38–39); 6% of women had a gestational age of less than 37 weeks and for 21% of women gestational age was not recorded on their client card and/or they did not know." Are you referring to gestational age age birth? Please clarify. Also, can you please provide more information around why you chose to present the median gestational age of 39 weeks, rather than the mean? I may make more sense to present the mean.	We have revised the text to make it clear that this was the age recorded at admission: "The median gestational age of women on admission was 39 weeks (IQR 38–39)" The median was selected as it better represents the distribution of the data than the mean.
10 In the results section, you state " Eleven percent of women were seen by a skilled birth attendant (doctor, midwife, nurse), 39% by a community health extension worker (CHEW) or junior CHEW and 50% by an 'other' birth attendant. 'Other' included traditional birth attendants, environmental health officers/technicians/assistants and health attendant/assistant." Are you able to identify at which state along the continuum of pregnancy these types of health workers were seen and at what point (if any) did the cadre of health worker change?	The study only looked at the period of childbirth and so we cannot comment on the state along the continuum of pregnancy. The cadre was recorded at the start of the observation period and we have updated the wording as follows to clarify: "At the start of the observation period, 11% of women were attended by a skilled birth attendant (doctor, midwife, nurse), 39% by a CHEW or junior CHEW and 50% by an 'other' birth attendant. 'Other' included environmental health officers/technicians/assistants (43%), health attendant/assistant (43%), traditional birth attendants (4%), community health officer (1%) and other (9%)." We are not able to determine if cadres changed as the observation progressed.
11 page 10 of 35 - please double check your terminology with respect to "panel 1". Is this terminology consistent with author guidelines for submission?	Could the editors please advise as we cannot find any guidance on this in the instructions for authors. Subsequent note: panel 1 has now been removed.
12 Page 11 of 35, lines 35 and 42. Please check your formatting	Thank you for your careful review.
13 In your methods section, please discuss how	We have now removed the composite score. See

Comment by reviewer 2	Response
your arrived at the 10 criteria used for the composite score. What made some criteria more desirable to capture than others? Please provide your rationale.	response to comment 2.
14 It may make sense to provide an abridged version of your supplementary table that corresponds to the 10 item composite score in your results section. It would also make sense for you to present the results in written form as they relate to the 10 item composite score.	We have now removed the composite score. See response to comment 2.
15 Page 14 of 35 - you speak to guidelines for midwives, nurses, and community health workers. Please ensure that you are discussing your results based on the cadre of health care worker that you have observed as part of your study.	The majority of health care workers observed in Gombe State would not be classified as skilled by international definition. We provide a more thorough explanation of the context, and also on page 14 line 352, we discuss the fact that many of the health care workers in this setting are excluded from training programmes.
16 After the results have been ammended, changes may need to be made to the discussion based on the reorganization of your results.	No further action required.
17 While Figures 1-5 are useful, it may make more sense to present this information in table format so that one may be able to see the trends over time with respect to each component.	A full table of results is presented in the supplementary material.